# Personal Passive Air Samplers for Chlorinated Gases Generated from the Use of Consumer Products

**DOI:** 10.3390/ijerph18178940

**Published:** 2021-08-25

**Authors:** Yeonjeong Ha, Yerim Koo, Jung-Hwan Kwon

**Affiliations:** Division of Environmental Science and Ecological Engineering, Korea University, 145 Anam-ro, Seongbuk-gu, Seoul 02841, Korea; gbhyjyh@korea.ac.kr (Y.H.); gsd516@korea.ac.kr (Y.K.)

**Keywords:** personal passive air sampler (PPAS), chlorinated disinfectant, chlorine gas, chamber test

## Abstract

Various chlorine-based disinfectants are being used during the COVID-19 pandemic; however, only a few studies on exposure to harmful gases resulting from the use of these disinfectants exist. Previously, we developed a personal passive air sampler (PPAS) to estimate the exposure level to chlorine gas while using chlorinated disinfectants. Herein, we investigated the color development of the passive sampler corresponding to chlorine exposure concentration and time, which allows the general population to easily estimate their gas exposure levels. The uptake and reaction rate of PPAS are also explained, and the maximum capacity of the sampler was determined as 1.8 mol of chlorine per unit volume (m^3^) of the passive sampler. Additionally, the effects of disinfectant types on the gas exposure level were successfully assessed using passive samplers deployed in a closed chamber. It is noteworthy that the same level of chlorine gas is generated from liquid household bleach regardless of dilution ratios, and we confirmed that the chlorine gas can diffuse out from a gel-type disinfectant. Considering that this PPAS reflects reactive gas removal, individual working patterns, and environmental conditions, this sampler can be successfully used to estimate personal exposure levels of chlorinated gases generated from disinfectants.

## 1. Introduction

Coronavirus disease 2019 (COVID-19), an ongoing global pandemic, had resulted in 170,426,245 confirmed cases, including 3,548,628 deaths worldwide as of 1 June 2021 (WHO Coronavirus (COVID-19) Dashboard, available at https://covid19.who.int/ (accessed on 1 June 2021)). To prevent the transmission of severe acute respiratory syndrome coronavirus 2 (SARS-CoV-2), which causes COVID-19, cleaning and disinfecting frequently touched surfaces in households and workplaces using disinfectants are highly recommended by the Centers for Disease Control and Prevention (CDC) [1] and the World Health Organization (WHO) [2]. With this current guideline, the demand for disinfectants has increased, resulting in a boosted supply of disinfectants against SARS-CoV-2 [3]. To prevent the improper use of unsafe products and accidental poisoning, the United States Environmental Protection Agency (EPA) and the Ministry of Environment (MOE) of Korea provided a list of approved disinfectants against SARS-CoV-2 with their active ingredients and guidelines for safe use [4,5].

The majority of registered disinfectants for COVID-19 have sodium hypochlorite (NaOCl) as an active ingredient [6]. A sodium hypochlorite solution, generally known as bleach, is an effective disinfectant against a broad range of microorganisms but has low toxicity toward humans [6]. Thus, bleach has been used in a broad range of applications, such as disinfecting surfaces, cleaning bathrooms, sanitizing swimming pools, and laundry bleaching. However, chlorinated gases, such as chlorine (Cl_2_), hypochlorous acid (HOCl), and chloramines, are generated when chlorinated disinfectants are used [7,8], and these chlorinated gases could be toxic via inhalation, causing diseases, such as asthma, respiratory issues, and lung injury [9,10]. The unsafe use of chlorinated disinfectants (e.g., mixing bleaches with acid solutions) generates high concentrations of chlorine gas, which causes reactive airway disinfection syndrome (RADS) [11,12]. Indeed, there was a sharp increase in the daily number of calls for exposure to disinfectants due to the COVID-19 pandemic in the United States, and bleaches and inhalation accounted for the largest percentage of the increase among all disinfectants and all exposure routes, respectively [13]. In addition, an internal panel survey conducted by the U.S. Department of Health and Human Services reported that many people do not know how to use the bleach solution safely [14], which results in accidental inhalation poisoning caused by chlorinated gases. Thus, it is imperative to estimate personal gas exposure from the use of chlorine-based disinfectants.

Previously, we developed a personal passive air sampler (PPAS) that can be used to effectively determine the exposure level to chlorine gas while using consumer products of chlorine disinfectants [15]. PPAS consists of polydimethylsiloxane (PDMS), a gas-permeable medium, and *o*-dianisidine, which is doped inside the PDMS sheets. *o*-Dianisidine is a redox dye that reacts with oxidizing gases, and it clearly changes its color from transparent to green after exposure to chlorinated gases, and the color intensity increases with increasing exposure concentrations and time. The detailed chemical reactions and the color changes after the reaction are shown in our previous study [15]. The PPAS-derived chlorine gas concentration can be calculated from the reduced amount of *o*-dianisidine by the oxidation reaction with the chlorinated gases and the sampling rate of PPAS determined in the previous study [15]. We also applied the passive samplers to a panel study to estimate the chlorine gas exposure concentration while cleaning residential bathrooms with spray and liquid types of household bleaches [15]. The PPAS-derived chlorine concentrations were determined as 69–408 ppbv (parts per billion by volume) and 148–435 ppbv with spray and liquid-type bleaches, respectively. The PPAS-derived exposure levels were similar to the direct measurements of chlorine gas concentrations generated from bleaches in previous studies [7,8].

Although the PPAS has been successfully developed and applied to a panel study to estimate the personal exposure level of chlorinated gases, color development of the passive samplers based on the exposure level to chlorinated gases was not assessed in detail. In addition, the effects of chlorinated disinfectant types (e.g., liquid, spray, and gel) and dilution of liquid bleach on exposure to chlorine gases were not evaluated. In previous studies, closed chambers were successfully used to calibrate the passive air samplers (e.g., determination of the sampling rate, gas uptake rates, and capacity of the samplers) because the closed chamber can minimize the effects of environmental factors, such as wind velocity and surface contaminations [16,17,18]. Thus, further assessment of the passive samplers deployed in the closed-chamber system is necessary to define the color changes according to the exposure concentrations and time, uptake and reaction rates, and maximum capacity of the samplers.

The objectives of this study were (1) to investigate the color development of the PPAS corresponding to the exposure concentrations and time, (2) to assess the uptake and reaction rate of the PPAS and determine its capacity; and (3) to investigate the effects of disinfectant type (e.g., liquid, spray, and gel) and dilution factors of liquid bleach on gas chlorine exposure.

## 2. Materials and Methods

### 2.1. Materials

PDMS was purchased from Specialty Silicone Products, Inc. (Ballston Spa, NY, USA). Liquid-type chlorinated disinfectants (4.00 *w*/*v* NaOCl) were obtained from Yuhanrox (Yuhan-clorox Inc., Seoul, Republic of Korea), and spray (~1.7% *w*/*v* NaOCl) and gel-type (~1.8% *w*/*v* NaOCl) disinfectants were the products of LG Household & Health Care, Ltd. (Seoul, Republic of Korea). *o*-Dianisidine and NaOCl solution (available chlorine 4.00–4.99%) was obtained from Sigma-Aldrich (St. Louis, MO, USA).

### 2.2. Personal Passive Air Sampler (PPAS) Preparation

The method for preparing the passive air sampler was described in our previous study [15]. In brief, PDMS sheets with 0.55-mm thickness were cut into 1.5 cm × 1.5 cm, and the PDMS pieces were cleaned using n-hexane and methanol. Then, PDMS was soaked in 0.5 g L^−1^ *o*-dianisidine dissolved in a toluene solution over 24 h, followed by removal of the PDMS from the solution and drying for 2 h in a fume hood. Any remaining *o*-dianisidine on the PDMS surface was cleaned with methanol. The passive sampler patches were prepared by attaching three individual PDMS sheets on a paper using a stapler, as shown in Figure 1b. All passive sampler patches were wrapped with Kimwipes and aluminum foils and stored in plastic bags sealed with vacuum packing until use to prevent any reaction of *o*-dianisidine with oxidizing gases in the ambient air.

### 2.3. Measurement of Chlorine Concentration Using Proton Transfer Reaction/Selective Reagent Ionization Mass Spectrometer (PTR/SRI-MS)

To measure the instantaneous concentration of chlorine gas in the chamber (C_PTR-MS, instataneous_), a proton transfer reaction/selective reagent ionization-mass spectrometer (PTR/SRI-MS, IONICON, Innsbruck, Austria) was employed with O_2_^+^ as the selective ion. The instantaneous chlorine concentration was calculated by summation of the PTR/SRI-MS raw response differences between the inside and outside chambers at m/z = 70 (^35^Cl^35^Cl^−^), 72 (^37^Cl^35^Cl^−^, ^35^Cl^37^Cl^−^), and 74 (^37^Cl^37^Cl^−^) and the Cl_2_ gas calibration curve of PTR/SRI-MS, which is described in detail in our previous study [15]. The time-weighted average concentration of chlorine gas (C_PTR-MS,TWA_, ppbv) was calculated using Equation (1):(1)$$C_{PTR-MS,TWA}=\frac{\int_{0}^{t}(C_{PTR-MS,instantaneous}\left(t\right)\times t)\;dt}{\int_{0}^{t}dt}$$
where *C_PTR-MS, instantaneous_* is an instantaneous chlorine concentration measured by PTR/SRI-MS, and *t* is the deployment time of the passive samplers in the chamber (h).

### 2.4. Chamber Study for Estimating Passive Sampler-Derived Chlorine Gas Equivalent Exposure

The chamber design is illustrated in Appendix A. A 125 L acryl chamber was used, and each amount of chlorinated disinfectant was spread or sprayed onto the bottom of the chamber. The passive sampler patches were hung on the top of the chamber and taken out of the chamber at the designated times. Then, each individual passive sampler was placed into 10 mL isopropyl alcohol to extract *o*-dianisidine from the sampler overnight while waiting for the UV/Vis measurement (instrument model). *o*-Dianisidine shows maximum absorbance at 305 nm from the UV/Vis measurement of extraction solvents. The oxidized form of *o*-dianisidine synthesized from the oxidation reaction of *o*-dianisidine with chlorine (i.e., dianisidine quinonediimine) has a maximum absorbance at 429 nm.

The chlorine mass reacted with *o*-dianisidine per unit volume of passive samplers (*C_s_*, mol m^−3^) was calculated as follows:(2)$$C_{S}=\left(C_{i}-C_{f}\right)\times V_{E}\times \frac{1}{M_{w}}\times \frac{1}{V_{sampler}}\times \frac{1}{2}$$
where *C_i_* and *C_f_* are the initial and final concentrations in the extraction solvents (g L^−1^), respectively; *V_E_* is the volume of the extraction solvent (isopropyl alcohol, 0.01 L); M_w_ is the molecular weight of *o*-dianisidine (244.30 g mol^−1^), and *V_sampler_* is the volume of the individual passive sampler (m^3^).

Then, the time-weighted average chlorine concentration in air obtained from the passive air sampler (*C_v,TWA_*, mol m^−3^) was calculated using Equation (3):(3)$$C_{v,TWA}=\frac{M_{sampled}}{R_{s}t}=\frac{C_{S}V_{sampler}}{R_{S}t}\;(3)$$
where *M_sampled_* is the mass of *o*-dianisidine inside the passive sampler that reacts with chlorine gas (mol), *R_s_* is the sampling rate (0.00253 m^3^ h^−1^), *t* is the deployment time of the passive sampler in the chamber (h), and *C**_S_* and *V_sampler_* are described in Equation (2). We obtained the sampling rate of the passive sampler using NaOCl solution (available chlorine 4.00–4.99%) spread on the bottom of the 125 L chamber, and more details for calibrating the sampling rate are described in our previous study [15].

## 3. Results and Discussions

### 3.1. Color Changes of the Passive Sampler According to Chlorine Exposure Concentration and Time

Instantaneous chlorine gas concentrations generated by spreading 20 mL of NaOCl solution (available chlorine 4.00–4.99%) at the bottom of the 125 L test chamber are shown in Figure 1a. The instantaneous chlorine concentrations were measured using a proton transfer reaction/selective reagent ionization-mass spectrometer (PTR/SRI-MS). The time-weighted average (TWA) chlorine concentrations were also calculated from instantaneous concentration measurements and are shown in Figure 1a. Zero min in Figure 1a indicates 2 h after spreading the NaOCl solution in the chamber and the time point when the passive samplers were deployed in the chamber. As shown in Figure 1a, the chlorine concentration continuously increased and reached 1160 ppbv at 120 min after passive sampler deployment, and the TWA chlorine concentrations ranged from 552 to 878 ppbv. The raw response of PTR/SRI-MS, observed ratios of chlorine isotopes, and instantaneous chlorine concentrations at each time point are presented in Appendix A.

Figure 1b shows the color changes of the passive samplers after 5–120 min of deployment in the 125-L chamber and the corresponding TWA chlorine concentration times and exposure time (CT, unit: ppbv min). Green color density significantly increases with increasing CT values, and the color changes can easily be distinguished by the naked eye. According to the occupational safety and health administration (OSHA), the permissible exposure limit of chlorine is 500 ppbv (time-weighted average) [19]. In addition, the acute exposure guideline level 1 (AEGL-1, end point: no to slight changes in pulmonary function parameters in humans) for chlorine was reported as 500 ppbv for 10 min–8 h of exposure [20]. Assuming that inhalation toxicity depends on CT values, Figure 1b shows that exposure times with constant 500 ppb chlorine exposure can be easily estimated by the color changes of the passive samplers. Thus, this passive sampler can be used as an indicator of low levels of chlorine gas exposure generated from consumer products of chlorinated disinfectants without any detection instruments.

### 3.2. Uptake and Reaction Rate of the Passive Samplers

The passive air samplers used in this study have two main processes: (i) chlorine mass flow from ambient air to the passive sampler and (ii) oxidation of chlorine molecules with *o*-dianisidines doped inside passive samplers (Figure 2a). Thus, the total mass flux (*F_total_*) can be determined by the mass flow rate of chlorine gas diffusion from air to the sampler (*F_diffusion_*), and the reaction rate of chlorine with *o*-dianisidine (*F_reaction_*). *F_diffusion_* can be calculated by Fick’s first law, and F_reaction_ can be determined using first-order reaction kinetics, assuming that the reaction is a first-order reaction. The total mass flux (*F_total_*) is:(4)$$F_{total}=V_{s}\frac{dC_{s}\left(t\right)}{dt}=\frac{1}{\frac{1}{F_{diffusion}}+\frac{1}{F_{reaction}}}=\frac{1}{\frac{1}{k_{o}A_{s}\left(C_{A}-\frac{C_{s}\left(t\right)}{K_{SA}}\right)}+\frac{1}{k_{reac}A_{S}C_{s}\left(t\right)\delta }}$$
where *C_S_* (*t*) is the chlorine concentration diffused inside the passive samplers (mol m^−3^), *t* is the passive sampler deployment (h), V_s_ is the volume of the passive sampler (m^3^), *k_O_* is the overall mass transfer coefficient including air and sampler sides (m h^−1^), *A_S_* is the surface area of the passive samplers (2.25 × 10^−4^ m^2^), C_A_ is the air-side chlorine concentration (mol m^−3^), K_SA_ is the partitioning constant of chlorine between the sampler and air, k_reac_ is the first-order reaction-rate constant for the redox reaction of chlorine with *o*-dianisidine (h^−1^), and *δ* is the thickness of the passive samplers (0.005 m).

In the early stage, since the reaction rate is much faster than the diffusion mass flux (*F_reaction_* ≫ *F_diffusion_*), and chlorine concentration in passive samplers are negligible (C_s_(t)~0), Equation (4) can be simplified to *F_total_* = *k_O_A_S_C_A_*. On the other hand, at the stage when chlorine gas inside the passive samplers reaches the maximum concentration that can react with *o*-dianisidine, the reaction rate is negligible compared with the diffusion flux (*F_reaction_* ≪ *F_diffusion_*), which results in *F_total_* ~ 0. Thus, theoretically, at the initial step, the chlorine mass reacted with *o*-dianisidine (*C_S_*) linearly increases with time, assuming that *C_A_* is constant, and finally, it reaches maximum capacity with no *C_S_* changes with time. The theoretical uptake and reaction curves of chlorine gas for the passive samplers are shown in Figure 2b.

Figure 2c shows the changes in *C_S_* of the passive samplers with time until equilibrium is reached and the effects of a fan on it. It is generally acknowledged that wind around passive samplers decreases the thickness of the air boundary layer, which allows faster diffusion of gas molecules and aerosols [21,22]. As shown in Figure 2b, the experimental reaction and diffusion curve (Figure 2c) follow the theoretical curve (Figure 2b) as well as the theoretical uptake profiles of the passive air samplers for volatile organic compounds (VOCs), semi-volatile organic compounds (SVOCs), and explosive vapors [23,24,25]. It is obvious that the chlorine mass transfer rate from air to the passive sampler is faster with a fan than without a fan. However, the maximum *C_S_* values reached approximately 1.8 mol m^−3^ regardless of using a fan, which corresponds to a concentration time (CT) value of 104,000 ppbv min at 3 h (the concentration can be calculated using Equation (4) below). Thus, the maximum capacity of the passive samplers is 1.8 mol m^−3^, which corresponds to an exposure level of 104,000 ppbv min. To apply passive samplers to CT values above 104,000 ppbv min, the samplers should be modified to increase the surface area.

### 3.3. Effects of Disinfectant Types on the Chlorine Gas Concentration

Liquid, spray, and gel types are the three most commonly used chlorine disinfectants. Herein, we spread a liquid-type disinfectant with different dilution factors, spray type with different usage amounts, and gel-type disinfectant in a 125-L chamber, which reacted with a redox dye in the passive samplers (*C_S_*) and time-weighted average concentration of chlorine gas (*C_w_*_,*TWA*_) (Figure 3).

Liquid household bleach typically contains 4–10% sodium hypochlorite (NaOCl), and there are various dilution ratios (from 25 to 500 times dilution) recommended for consumers depending on the purpose of the bleach use. It is generally believed that more diluted liquid bleach can generate a lower chlorine gas concentration. However, as seen in Figure 3a,d, the chlorine gas concentration generated from the original bleach (liquid bleach without dilution) was slightly higher than the gas concentrations from bleach with 50 times and 100 times dilution, and there was no significant difference between the concentration of chlorine gas generated from the 50 times dilution and 100 times dilution bleach. The formation of chlorine gas from a sodium hypochlorite solution strongly depends on the pH of the solution. Undiluted bleach is a strong alkaline solution (11 < pH <13) that minimizes chlorine gas generation. However, once the bleach was diluted, the pH of the solution decreased, generating more chlorine gas. The pH of the undiluted bleach in this study was 12.1, and the 50 times dilution and 100 times dilution bleach had pH values of 10.4 and 10.1, respectively. These decreases in pH can overcome the decrease in total chlorine concentration in bleach by dilution, creating the same level of chlorine gas. With 200 mL of original and diluted liquid bleach, C_s_ values linearly increased with time, and the time-weighted average air concentration of Cl_2_ gas after 2 h ranged from 300 to 500 ppbv.

With a spray-type chlorinated disinfectant (~1.7% NaOCl), C_s_, C_v_, and TWA were significantly affected by the total mass of use (Figure 3b,e). When the spray was used one and five times, *C_S_* linearly increased with time until 3 h, and when the spray was used 10 times, *C_S_* values reached the maximum capacity. The average *C_v_*_,*TWA*_ values after 1 h when the spray was used 10 times, 5 times, and 1 time were 947 ppbv, 756 ppbv, and 204 ppbv, respectively. Even though the NaOCl content in the spray disinfectant (~1.7% NaOCl) was significantly lower than that in the liquid-type disinfectant (4.00% NaOCl), the chlorine gas concentration level created by the spray-type disinfectants was higher or similar to that of the liquid bleach, indicating that gas compounds easily diffused out from the small droplets of the spray-type disinfectants.

Finally, a gel-type disinfectant (50.80 g) containing 1.8% sodium hypochlorite was used. As shown in Figure 3c, *C_S_* increased with time, with a relatively lower linear increasing rate at the beginning (from 0 to 1.5 h) and a higher increasing rate after 1.5 h. This result implies that components in the gel-type product (e.g., thickener) prevent the diffusion of gases from the product at the beginning. The absorbance of the extraction solution of the passive sampler at 429 nm, which represents the oxidized form of *o*-dianisidine, clearly shows that a small amount of oxidized *o*-dianisidine was synthesized, but a significant amount of oxidized *o*-dianisidine was synthesized after 1.5 h (Appendix A). The average *C_v_*_,*TWA*_ values ranged from 215 to 301 ppbv (Figure 3f), suggesting that the level of chlorine gas exposure after using a gel-type disinfectant can be similar to the exposure level from liquid bleach, especially those used in enclosed chambers without ventilation.

## 4. Conclusions

The passive air samplers for detecting chlorine gas while using chlorinated disinfectants effectively developed the color in the TWA chlorine concentrations ranging from 552 to 878 ppbv. We also investigated the uptake and reaction rate of the passive samplers, indicating that the maximum capacity of the samplers is 1.8 mol m^−3^. In addition, the samplers were successfully applied for estimating the level of chlorine gas concentrations generated from different types (e.g., liquid, spray, and gel) of chlorinated disinfectants.

The personal wearable passive air sampler assessed in this study can reflect the working environment (e.g., temperature, humidity, indoor and outdoor lights, surface contamination, and indoor air quality) and personal working patterns, which are critical factors for estimating exposure to reactive gases. In addition, this brooch-type passive sampler attached near the breathing zone can properly approximate inhalation exposure, as reported in previous studies [26,27]. Furthermore, this passive sampler can prevent accidental inhalation poisoning caused by chlorinated gases because color changes of the samplers can be easily detected with the naked eye.

Considering that the active chlorine form in chlorinated disinfectants varies (e.g., HOCl, NaOCl, Ca(OCl)_2_, and ClO_2_), and oxidizing gases generated from the use of these disinfectants are even more diverse, future work is required to test the reactivity of the passive samplers with different chlorinated gases. In addition, using passive samplers, further experiments can be conducted to confirm the importance of indoor environments and personal working patterns on inhalation exposure.

## Figures and Tables

**Figure 1 ijerph-18-08940-f001:**
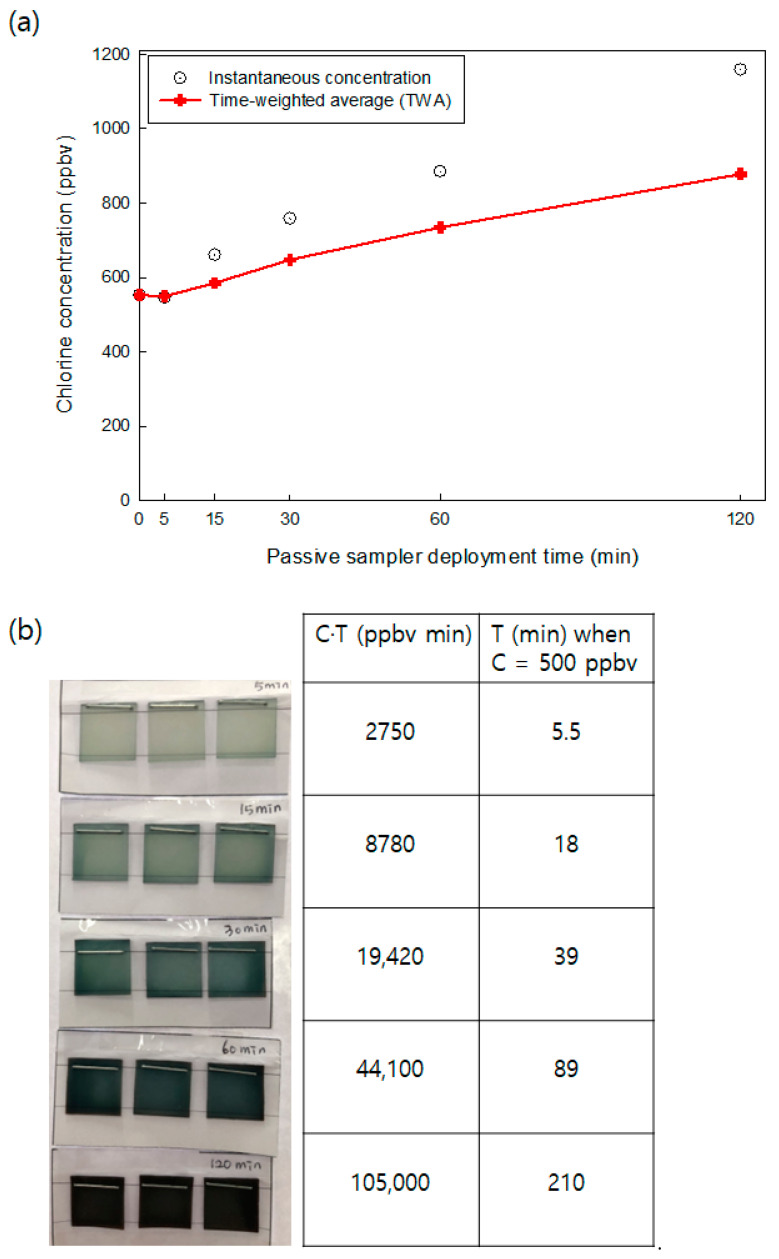
(**a**) Instantaneous chlorine concentration and time-weighted average (TWA) chlorine concentration (ppbv) in the 125 L chamber after spreading 20 mL NaOCl (available chlorine, 4.00–4.99%) solution. The chlorine concentrations were measured by PTR-MS. (**b**) Color change of passive samplers after 5–120 min of passive sampler deployment in the 125 L chamber. The table shows CT (TWA chlorine concentration × time, unit: ppbv min) at each time and the exposure time when chlorine concentration was assumed to be 500 ppbv, with permissible exposure limits of chlorine reported by the Occupational Safety and Health Administration (OSHA) and acute exposure guideline 1 (AEGL-1).

**Figure 2 ijerph-18-08940-f002:**
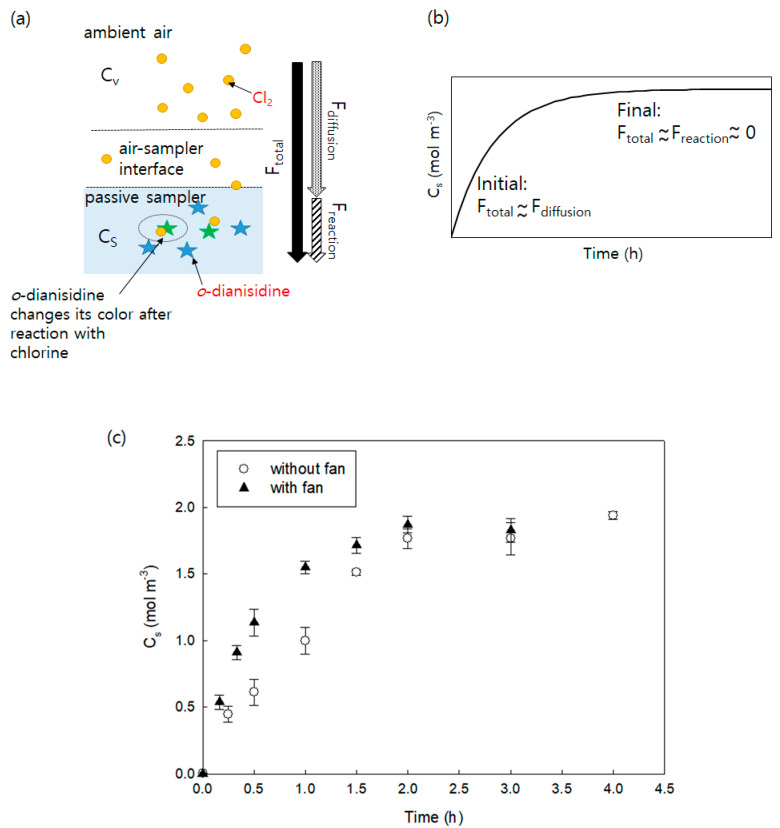
(**a**) Diffusion and reaction scheme of the passive sampler and (**b**) theoretical uptake and reaction curve of chlorine gas for the passive sampler. (**c**) Changes of C_s_ (chlorine mass reacted with *o*-dianisidine per unit volume of passive samplers) with time. Error bars represent the standard deviations of triplicates.

**Figure 3 ijerph-18-08940-f003:**
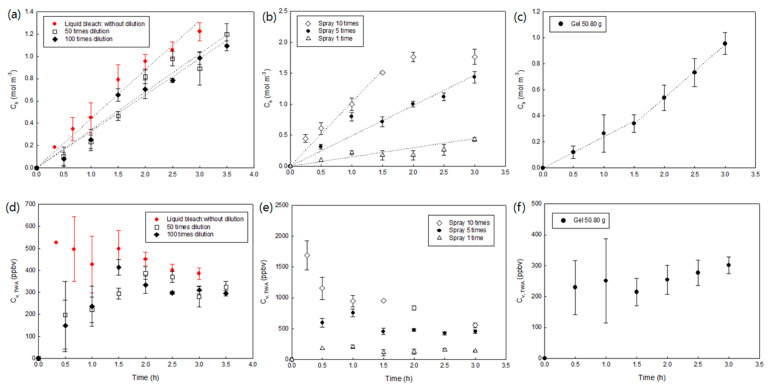
Effects of the type of chlorine-based disinfectants on C_s_ (chlorine mass reacted with *o*-dianisidine per unit volume of passive samplers) and C_v,TWA_ (time-weighted average chlorine concentration in air obtained from passive samplers). (**a**–**c**) shows how C_s_ changes with time using liquid bleach, spray-type, and gel-type chlorinated disinfectants, respectively. Dotted lines in (**a**–**c**) are linear regressions applied in the range that C_s_ increased linearly with time. (**d**–**f**) shows C_v,TWA_ using liquid bleach, spray-type, and gel-type chlorinated disinfectants, respectively. Error bars represent the standard deviations of triplicates.

## Data Availability

The data presented in this study are available on request.

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
