# Peer review of "Personal Passive Air Samplers for Chlorinated Gases Generated from the Use of Consumer Products"

_ijerph, 2021, doi:10.3390/ijerph18178940_

Round 1
Reviewer 1 Report
- This manuscript is well written and reports the evaluation of a personal passive air sampler for chlorinated gases generated from the use of consumer products. It has been tested with regard to color development corresponding to the exposure concentrations and time, the uptake and reaction rate, the PPAS’s capacity; and the effects of disinfectant type as well as dilution factors.
- In general, I think appropriate experiments have been described and interpreted in order to evaluate the feasibility of the approach.
- Only a few minor revisions are required before this article can be accepted for publication.
- I was wondering how specific the detection is with regard to chlorinated gases. Are there any known side reactions of o-dianisidine and what kind of components in the work place environment might influence its performance. In general, I would recommend applying it to a "real world" application outside of a laboratory chamber for future studies.
Results and discussions
- Line 227: replace ‘Effects pf dosomfectat’ by ‘Effects of dosomfectat’’
- Section 3.2.: Figure 2(a) is not mentioned in the text section. I would recommend to implement a description of Figure 2(a) in the text.
Author Response
Dear Reviewer:
We thank the reviewer for considerate comments and constructive suggestions for improvement of the manuscript. We delineate below all answers and changes made in response to the reviewer’s comment.
-. I was wondering how specific the detection is with regard to chlorinated gases. Are there any known side reactions of o-dianisidine and what kind of components in the work place environment might influence its performance.
Response: o-Dianisidine is a redox dye that changes its color by the oxidation reaction. Thus, o-dianisidine can react with any oxidizing gases, not only chlorinated gases but also oxygen in the ambient air. However, we checked that when the passive sampler exposed to the ambient air, the color of passive sampler changed from transparent to brown and the reaction rate is very slow compared with its reaction with chlorinated gases.
-. In general, I would recommend applying it to a "real world" application outside of a laboratory chamber for future studies.
Response: Thank you for the suggestion. In our previous study (Ha et al., 2021), we tested the passive air samplers with the help of ten volunteers and estimated the personal exposure level of chlorine equivalent gas after bathrooms were cleaned using household disinfection products. In addition, we have a plan to apply the passive samplers to estimate chlorinated gas exposure level while cleaning and disinfecting the public places at the end of this year.
Ha et al., 2021, Development of a personal passive air sampler for estimating exposure to effective chlorine while using chlorine-based disinfectants. Indoor Air 31, 557-565.
-. Line 227: replace ‘Effects pf dosomfectat’ by ‘Effects of dosomfectat’’
Response: Thank you for finding typos. We corrected the subtitle as “Effects of disinfectant types on the chlorine gas concentration”
-.Section 3.2.: Figure 2(a) is not mentioned in the text section. I would recommend to implement a description of Figure 2(a) in the text.
Response: We mentioned Figure 2(a) in the revised manuscript (Line 187).

Reviewer 2 Report
Congratulations! I have only recommended minor corrections.
General comments: this article studies the use of a special sampler chamber to measure remained concentrations of chlorine derivatives from cleaning products. In general, the whole article is very well organized and written. Tables and figures are adequate and very well described. Conclusions are also well addressed. I wonder if this system could be applied to other gases/liquids used against COVID as hydroalcoholic mixtures. I only recommend some minor changes to improve it.
Abstract:
Lines
- Reactive gases is general, just report the Chlorine gases as these gases were the ones
analysed.
- Introduction:
Lines
68 The authors should explain what are the units ppbv. Take into account that ppb are not
meaning the same units in USA than in Europe.
In this section a miss a part reporting more in detail the reactions of chlorine species and
also the reaction with o-dianisine.
- Materials and Methods
Lines
117 Terms in equation 1 must be described here.
- Results
Lines
185 Why the authors calculate the parameters considering a first order kinetics?
272 Why standard deviations are so high for times < 1.5 hours (Figure 3)?
References:
Cited Journals should be written in caps; for example: Food and Chemical Toxicology instead Food and chemical toxicology.
DOIs should be added to the cites.
No misprints neither format errors were detected
Author Response
Dear Reviewer:
We thank the reviewer for considerate comments for improvement of the manuscript. We delineate below all answers and changes made in response to the reviewer’s comment.
-. Abstract: Reactive gases is general, just report the chlorine gases as these gases were the ones analyzed.
Response: Following reviewer’s comment, we changed “reactive gases” to “chlorinated gases” in the revised abstract (Line 22).
-. Introduction: (Line 68) The authors should explain what are the units ppbv. Take into account that ppb are not meaning the same units in USA than in Europe.
Response: Following reviewer’s comment, we define the units ppbv (parts per billion by volume) in the revised manuscript (Line 71).
-. Introduction: In this section, a miss a part reporting more in detail the reactions of chlorine species and also the reaction with o-dianisidine.
Response: We explained the o-dianisidine reaction with oxidizing gases and cited our previous study that shows o-dianisidine’s detailed chemical reactions and the color changes (Line 61-65).
-. Materials and Methods: (Line 117) Terms in equation 1 must be described here.
Response: We defined the terms in equation 1 in Line 122-123 in the revised manuscript.
-. Results: (Line 185) Why the authors calculate the parameters considering a first order kinetics?
Response: Although we could not determine the reaction rate of o-dianiside and chlorine reaction, a few previous studies (Bishop and Hartshorn, 1971, Santana et al., 2019) have reported that the oxidation reactions of redox dye including benzidines generally followed a first-order kinetic.
Santana et al., 2019, Kinetic evaluation of dye decolorization by fenton processes in the presence of 3-hydroxyanthranilic acid. International Journal of Environmental Research and Public Health. 16, 1602.
Bishop and Hartshorn, 1971, Some observations on oxidation-reduction indicators of the benzidine, naphthidine and diarylamine types, Analyst, 66, 26-36.
-. Results: (Line 272) Why standard deviations are so high for times < 1.5 hours (Figure 3)?
Response: Cv,TWA values in Figure 3 were calculated based on the equation (3) in the manuscript ( ). In equation (3), Vsampler and Rs are constant values, and the standard deviation of Cv,TWA is equal to the standard deviation of Cs divided by time t. Thus, if the standard deviations of Cs are similar for all time measurements, the standard deviations of Cv,TWA increase as time t increases.
-. References: Cited journals should be written in caps; for example: Food and Chemical Toxicology instead Food and chemical toxicology; DOIs should be added to the cites.
Response: We changes the references following the reviewer’s comment (Cited journals are written in caps and DOIs are added to the references in the revised manuscript.)
